# Balancing resistor-based online electrochemical impedance spectroscopy in battery systems: opportunities and limitations

Alexander Blömeke [1,2,3] ✉, Hendrik Zappen[4], Florian Ringbeck [1,2,3], Fabian Frie [1,2,3], David Wasylowski [1,2,3] & Dirk Uwe Sauer [1,2,3,5] ✉

Active dissipative balancing systems are essential in battery systems, particularly for compensating the leakage current differences in battery cells. This study focuses on using balancing resistors to stimulate battery cells for impedance measurement. The value of impedance spectroscopy for in-depth battery cell diagnostics, such as temperature or aging, is currently being demonstrated and recognized by vehicle manufacturers, chip producers, and academia. Our research systematically explores the feasibility of using existing balancing resistors in battery management systems and identifies potential limitations. Here we propose a formula to minimize hardware requirements through signal processing techniques. A quadrupling of the sampling rate, number of averaging values, or the size of the fast Fourier transform is equivalent, concerning the signal-to-noise ratio, to increasing the analog resolution by one bit or reducing the input filter bandwidth by a quarter.

A battery management system (BMS) cover various functionalities in battery systems. Apart from other features, it provides battery state estimation, contactors control, and the measurement of physical cell states such as voltage, current, and temperature. Integral to a BMS is its ability to balance battery cells, which aims to achieve uniform state-of-charges (SoCs) across cells[1]. Presently, there are two main approaches to balancing: dissipative and non-dissipative systems.

Most commercial systems analyzed by the authors use dissipative balancing on the cell level. In dissipative balancing systems, a resistor transforms electrical energy into heat to equalize the SoC. It should be mentioned that such systems only compensate for deviations in the SoC. On the other hand, non-dissipative systems can compensate for SoC and state-of-health differences at the cell level. When comparing the two, dissipative systems offer advantages regarding reduced installation space and costs[2].

This study focuses on active dissipative balancing systems, aiming to analyze the implications of various components, such as the charging system, wiring, heat generation, and the leakage of batteries. The focus of this study is on the resistance of the balancing resistor. A resistor with small resistance can draw large currents from the battery during the equalization process. This current can be modulated to produce sinusoidal signals.

Electrochemical impedance spectroscopy (EIS) is used to characterize and track batteries' inner state by applying sinusoidal signals to the battery[3]. A decent signal-to-noise ratio (SNR) of the current and the voltage is necessary for a successful EIS measurement[4]. If the balancing current is high enough and the voltage measurement precise enough, the balancing current can be used for EIS measurements. The signal measurement's precision and the resistance must match to perform EIS measurements with an active dissipative balancing system. In the present work, it is hypothesized that, with a high balancing current and precise voltage measurements, active dissipative balancing systems can be used for EIS measurements.

To support the theoretical analyses, a demonstrator was built to test the feasibility of EIS measurements using an active dissipative balancing system. Our hands-on experiments provided useful insights that revealed both the potential and limitations of the approach. Based on our measurements and

[1]Chair for Electrochemical Energy Conversion and Storage Systems (ESS) - Institute for Power Electronics and Electrical Drives (ISEA), RWTH Aachen University, Campus-Boulevard 89, 52074 Aachen, Germany. [2]Center for Ageing, Reliability and Lifetime Prediction of Electrochemical and Power Electronic Systems (CARL), Aachen, Germany. [3]Jülich Aachen Research Alliance - JARA-Energy, Aachen, Germany. [4]Safion GmbH, Tempelhofer Str. 12, 52068 Aachen, Germany. [5]IEK-12, Helmholtz Institute Münster (HI MS) - Forschungszentrum Jülich, Jülich, Germany. ✉e-mail: alexander.bloemeke@rwth-aachen.de; batteries@isea.rwth-aachen.de

comparison with a theoretical framework, we present optimization strategies supported by comprehensive SNR estimates. In addition, we dive into the signal processing domain to overcome the limitations of an analog-to-digital converter (ADC).

## Methods

### Sizing of an active dissipative balancing system

The resistance of the balancing resistor depends on different parameters of a battery system. The resistor itself, the wires connecting the resistor to the battery, the leakage of the battery, the charger, the voltage measurements, and others affect the balancing performance of the system.

The root cause of the need for a balancing system is the difference in SoCs caused by different self-discharge rates of the batteries. Equalization of the SoC becomes necessary during charging, as reaching the upper voltage limit of one cell limits or even stops the charging process. The resistance could be selected by a simple design that compensates for the smallest possible charge current. The balancing current has to pass the sensing wires. Thus, the wires' length and diameter can also limit the current. A lower resistor and wire resistance increase the highest possible balancing current. The highest possible current, together with the cell voltage, leads to the power that is dissipated. In most applications, an upper limit for the temperature of the balancing resistor is given. In addition, the resistance can be influenced by the operating strategy. The longer balancing is possible, the lower the balancing current can be.

**Battery self-discharge rate and operating strategy.** Normalization is necessary to compare the size of balancing resistors $R_{bal}$ between different systems. Normalizing the resistance to the capacity of the batteries in parallel is one option. Using the capacity in Ah, the capacity in Wh, or the average or maximum voltage for normalization is also feasible. The self-discharge increases with the specific surface area of the anode and cathode[5]. Thus, the capacity is not perfect for normalization because the specific surface area is not included in the capacity. This is important because high-energy cells have a higher coating thickness than high-power batteries[6].

A normalization including the voltage would be reasonable as e.g., ref. [7] showed that the self-discharge correlates with the voltage for lithium-nickel-cobalt-aluminum-oxide cells. It needs to be investigated if this correlation is also valid for other chemistries, e.g., lithium-nickel-manganese-cobalt, lithium-iron-phosphate, or sodium-ion batteries. The current through the resistor depends on the voltage and thus differs for different chemistries. This leads to the point that a comparison of balancing resistor sizes between, e.g., lithium-nickel-manganese-cobalt and lithium-iron-phosphate could only be possible if, e.g., also the voltage is included. As the following analysis is based on reverse engineering and no approach would be perfect, we use for now the capacity in Ah for nominalization. Furthermore, we assume that the cell design parameters, like coating thickness, keep the same when scaling the capacity.

Normalizing the resistance of the balancing resistor by the capacity can be done as follows. The Tesla Model S, as an example, had a balancing resistor resistance of $R = \frac{158\,\Omega}{4} = 39.5\,\Omega$ for a battery with a nominal capacity of $C_N = 3.1\,A\,h \cdot 74 = 229.4\,A\,h$[8]. These parameters can be normalized to a balancing resistance $R'_{bal}$ of:

$$R'_{bal} = R \cdot C_N = 9061.3\,\Omega A\,h. \tag{1}$$

This normalized balancing resistance can now be used to calculate the balancing resistor of an equivalent battery with a capacity of, e.g., $C_{N_{new}} = 100\,A\,h$:

$$R_{bal,new} = \frac{R'_{bal}}{C_{N_{new}}} = \frac{9061.3\,\Omega A\,h}{100\,A\,h} = 90.61\,\Omega. \tag{2}$$

Now, an example is calculated from scratch. Assuming that a battery system to be designed contains one cell showing no self-discharge and another cell

showing a very high self-discharge. This could happen if a cell or a module is replaced in a rather old system. This aims to be a worst-case estimate. The self-discharge also differs if a few cells of the pack are exposed to high temperatures for a short time[9]. The self-discharge rate can be measured by float currents or the open-circuit voltage. Only a few reports on measuring the float-current exist so far.

In ref. [7], a self-discharge of a 2.5 A h battery with approximately 300 μA is measured at 60 °C at 4.1 V. This condition showed the highest discharge in that paper. Normalizing the self-discharge to the nominal capacity leads to approximately 120 μA A$^{-1}$ h$^{-1}$. Based on this value, the balancing resistor size for the pack to be designed can be calculated. The battery system to be designed should have the following specifications:

$$C_{N,Pack} \overset{!}{=} 100\,kW\,h$$

$$V_{Pack} \overset{!}{\approx} 400\,V$$

$$\text{Cell Chemistry} \overset{!}{=} \text{NMC, with } V_{Cell,nom} = 3.7\,V \tag{3}$$

$$\approx 100 \text{ cells in series}$$

$$C_{N,Cell} = \frac{100\,kW\,h}{100} = 1000\,W\,h \approx 270\,A\,h.$$

Scaling the leakage delta to this system leads to: $I_{Leakage\ Cell} \approx 120\,μA\,A^{-1}\,h^{-1} \cdot 270\,A\,h \approx 32.4\,mA$. This is equivalent to $24\,h \cdot \frac{32.4\,mA}{270\,A\,h} \cdot 100\,\% \approx 0.288\,\%$ per day or $\approx 8.64\%$ per month. This is higher than $< 5\,\%$ per month summarized in ref. [10] or ref. [9].

Two further parameters impact the resistance of the balancing resistor. First, the proportion of time when the application is balancing. Second, the lowest voltage is where the self-discharge should be compensated by balancing. For the example, it is assumed that a minimum balancing voltage of 3 V is required. Furthermore, the balancing is assumed to be active in 1 of 48 h ($\approx 20.83\,‰$). This would lead to a balancing resistor with a resistance of:

$$R = \frac{3\,V}{32.4\,mA} \cdot \frac{1}{48} \approx 1.93\,\Omega. \tag{4}$$

Normalizing the resistance to the capacity $R'_{bal} = 1.93\,\Omega \cdot 270\,A\,h = 521.1\,\Omega A\,h$ shows that this is low compared to Equation (1) of the Tesla Model S.

Other values can be calculated or read from Fig. 1. The resistance in the figure is calculated for a minimum balancing voltage of 3 V. On the $y$-axis, the required leakage current to be compensated is given. The balancing time ratio per mille is on the $x$-axis. Together, they define the required resistance. Other ways of representation are possible. E-Q diagrams, as in ref. [11], are very suitable.

**Charging system.** Dissipative balancing aims to discharge the batteries with the highest voltage or SoC until the pack is equalized. This becomes necessary if the charging process is limited to a cell in series reaching the upper voltage or SoC limit. At the moment one cell reaches that limit, this cell and its balancing resistor limit the current the charger is allowed to deliver. If the upper voltage or SoC limit is reached for one cell, the power of the balancing resistor in parallel defines the upper charge power limit. If this is lower than the lowest power of the charger, the charging has to stop.

One approach to select the balancing resistor could be to align it with the lowest possible charging power the charger can supply. The producer can align the balancing resistor if the charger is designed for the product. If the device, e.g., a battery electric vehicle, depends on public charging infrastructure, the limit is given by the charging standard. In Europe, this is at the moment the combined charging system high power charging 350 charging standard, which has the highest lowest current of 5 A[12]. For a voltage level of 4.2 V, this would require a resistance of 840 mΩ and a power

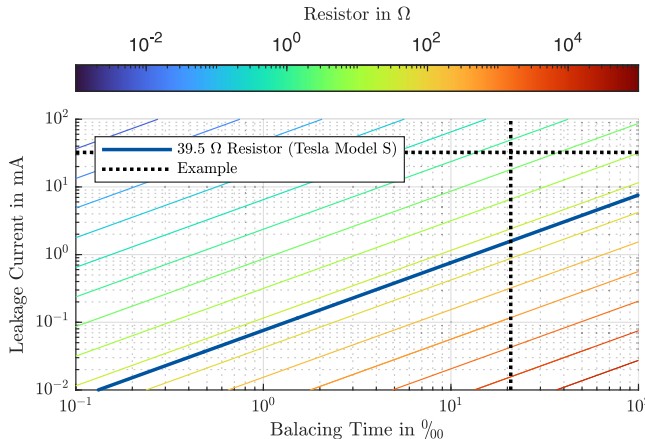

**Fig. 1 | Sizing of the active dissipative balancing resistor.** Suggested balancing resistor size for a balancing voltage of 3 V. The example assumes $\approx 20.83‰$ balancing time and a delta leakage current of 32.4 mA. The intersection of the two dashed lines leads to a resistance of 1.93 Ω.

dissipation of 21 W. Again, compared to the Tesla Model S in Equation (1), this would be a very low resistance.

**Thermal limitations and aging of resistors**. A balancing resistor in a dissipative balancing system is a secondary load used to discharge battery cells with too-high SoCs by converting electrical energy into thermal energy. Different SoCs can be adjusted faster with a smaller resistance. A smaller resistance leads to faster balancing but also higher power losses. Higher power losses lead to more heat. A proper thermal coupling to the cooling system becomes necessary. If the resistor is close to the battery, the battery's limitations must be considered.

High power losses lead to thermal cycling with a higher delta in temperature. Thermal cycling leads to mechanical stress[13] of the components. The mechanical stress could result in worse thermal conductivity[14] between the printed circuit board (PCB) and the resistor, thus increasing the maximum temperature again. Reliability has to be considered besides the thermal limitations of adjacent components and the overall system[15]. During the system design phase, an analysis of the thermal design and reliability of the components and the system forms a lower boundary of the resistance of the balancing system.

**Sense wire sizing**. The balancing resistance is not only the actual resistor as a discrete component but also the connectors and wires to the battery. The sensing wires' diameter, or the PCB trace cross-section, directly correlates with a voltage drop along the sensing wires. The voltage drop leads to a lower measured voltage, which could be calculated and corrected. To compare the resistance through the cable, the ideal resistance of a 1 m wire with $0.05\,\text{mm}^2 \approx A_{\text{AWG 30}}$ cross section is calculated:

$$R_{1\,\text{m},\,0.05\,\text{mm}^2} = \frac{\rho \cdot l}{A} = \frac{\rho \cdot 1\,\text{m}}{0.05\,\text{mm}^2}$$
$$= \begin{cases} 335.60\,\text{m}\Omega, & \rho_{\text{Copper},\,20\,°\text{C}} = 16.78\,\text{n}\Omega\text{m} \\ 530.00\,\text{m}\Omega, & \rho_{\text{Aluminum},\,20°\text{C}} = 26.5\,\text{n}\Omega\text{m} \end{cases} \quad (5)$$

The maximum balancing current of the resistor from Equation (1) is $\frac{4.2\,\text{V}}{39.5\,\Omega} \approx 106\,\text{mA}$. This current leads to a voltage drop of $530\,\text{m}\Omega \cdot 106\,\text{mA} \approx 56\,\text{mV}$ for aluminum, equivalent to a 1.34 % offset at 4.2 V.

For a battery like the lithium-iron-phosphate CATL 271 Ah[16] with $|\underline{Z}(1\,\text{kHz})| = 0.14 \pm 0.05\,\text{m}\Omega$, the added resistance by the wire is $\frac{530\,\text{m}\Omega}{0.14\,\text{m}\Omega} \approx 3786$ times higher than the resistance of the battery. This example

indicates that the wires' resistance may dominate a battery's resistance measurement.

Four-terminal pair[17] connections should be used to measure low resistances. This increases weight, installation space, and thus costs in an application. It depends on the application whether these disadvantages justify the gain of precision.

**Online electrochemical impedance spectroscopy**
Online EIS can be used for real-time battery monitoring[3]. Measuring the impedance allows early detection of critical states, such as temperatures[18,19] or plating during fast charging[20]. The additionally available data can be utilized to detect inhomogeneities in serial connections[21]. Furthermore, aging can be analyzed [22] to understand the degradation state. Implementing the evaluation of the impedance on an embedded system is a challenge[23]. The impedance is typically measured in the range from 10 mHz to 10 kHz. Nevertheless, higher frequencies can also be used to track and analyze batteries[24,25].

**Impedance calculation**. Several things are required in the battery system to allow online EIS. On the one hand, there needs to be a periodically available excitation of the battery containing frequencies suitable for battery diagnostics. On the other hand, a measurement setup with a decent resolution in both time and analog dimensions is necessary.

Based on Ohm's law[26], the impedance is the ratio of voltage and current, and if evaluated at multiple frequencies, it forms an impedance spectrum: $\underline{Z}(f) = \frac{\mathcal{F}(v(t))}{\mathcal{F}(i(t))}$. Both voltage $v(t) \in \mathbb{R}$ and current $i(t) \in \mathbb{R}$ are real-valued signals. Thus their Fourier Transforms are Hermitian. Therefore, the calculated impedance is Hermitian as well. In a BMS, the real-valued fast Fourier transform of libraries like common microcontroller software interface standard[27] can be used for the calculation.

A suitable excitation with a decent power at each measured frequency must be applied as mentioned. Based on Parseval's Theorem, single-tone sinusoidal excitation results in minimal energy in the time domain and maximal energy in the frequency domain[28]. For broadband analysis, ref. 4 gives an overview and shows that multisine signals show better SNR than chirp signals. Typically, an excitation is chosen for batteries, resulting in a voltage response of 10 $\text{mV}_{\text{p-p}}$[29,30].

**Signal-to-noise ratio in a battery management system**. In all battery applications, noise and distortions to impedance measurements decrease the quality of impedance measurements. The signal-to-noise and distortion ratio is a general measure for this purpose, which combines the SNR and further distortions.

$$\text{SINAD} = \frac{P_{\text{Signal}} + P_{\text{Noise}} + P_{\text{Distortion}}}{P_{\text{Noise}} + P_{\text{Distortion}}} \geq 1 \quad (6)$$

$$\text{SNR} = \frac{P_{\text{Signal}}}{P_{\text{Noise}}} \quad (7)$$

However, the distortion is strongly dependent on the application. Here, we focus on the SNR and formulas that can be generally applied. This omits, for example, the formula for pink noise.

In the following, clipping is ignored. Thus, the highest measured value is assumed to be below the full-scale amplitude range of the ADC. Assuming a perfect linear ADC with an full-scale amplitude range of $A_{\text{FSR}}$ and $N$ binary quantization steps leads to $\Delta A = \frac{A_{\text{FSR}}}{2^N}$ per quantization step. The signal has a root-mean-square of $A_{\text{RMS}}$, leading to an effective noise of:

$$P_{\text{Quantization Noise}} = \frac{\Delta A}{\sqrt{12}} = \frac{\frac{A_{\text{FSR}}}{2^N}}{\sqrt{12}} = \frac{A_{\text{FSR}}}{2^N \sqrt{12}}. \quad (8)$$

This can be combined and rewritten with the maximum absolute amplitude $|A_{max}|$ and the crest factor CF $\text{CF} = \frac{|A_{max}|}{A_{RMS}}$ to:

$$\begin{aligned}
\text{SNR}_{\text{ADC}_{\text{linear}},\text{dB}} &= 20\log_{10}\left(\frac{A_{RMS}}{\frac{A_{FSR}}{2^N\sqrt{12}}}\right) = 20\log_{10}\left(2^N\sqrt{12}\frac{A_{RMS}}{A_{FSR}}\right) \\
&= 20\log_{10}\left(2^N\right) + 20\log_{10}\left(\sqrt{12}\frac{A_{RMS}}{A_{FSR}}\right) \\
&= 20\log_{10}\left(2^N\right) + 20\log_{10}\left(\sqrt{12}\frac{\frac{|A_{max}|}{CF}}{A_{FSR}}\right) \\
&= 20\log_{10}\left(2^N\right) + 20\log_{10}\left(\sqrt{12}\frac{|A_{max}|}{A_{FSR}}\right) - 20\log_{10}(\text{CF}).
\end{aligned}$$
(9)

Equation (9) shows that reducing the crest factor CF will improve the SNR. Noise-like excitation signals lead to a low SNR. For single-sine zero mean signals with $A_{RMS} = \frac{A_{P-p}}{2\sqrt{2}} = \frac{A}{\sqrt{2}}$ this leads to the rule of thumb "6 dB per bit":

$$\begin{aligned}
\text{SNR}_{\text{ADC}_{\text{linear}},\text{dB}} &= 20\log_{10}\left(2^N\right) + 20\log_{10}\left(\sqrt{12}\frac{\frac{A_{p-p}}{2\sqrt{2}}}{A_{FSR}}\right) \\
&= \underbrace{20\log_{10}\left(2^N\right)}_{\approx 6.02\,\text{dB}\cdot N} + \underbrace{20\log_{10}\left(\sqrt{\frac{3}{2}}\right)}_{\approx 1.76\,\text{dB}} + \underbrace{20\log_{10}\left(\frac{A_{p-p}}{A_{FSR}}\right)}_{\substack{A_{p-p}=A_{FSR} \\ =0\,\text{dB}}}.
\end{aligned}$$
(10)

**Distortions in a battery management system.** Besides the ADC performance, other distortions can impact or even limit the SNR of the measurement. The number format can be a limitation, especially on an embedded system. For fixed-point formats, like the Q7, Q15, and Q31 on ARM controllers, the SNR can be calculated by $20\log_{10}\left(2^N\right)$, where in this case, N is equal to the number behind Q. For floating-point arithmetic according to the IEEE 754[31] the SNR can be calculated by $10\log_{10}(5.55) + 20\log_{10}\left(2^N\right)$ where N is now the bits of significant together with the hidden leading bit[32,33].

Partly caused by the ADC but also affected by other components of the measurement chain, the jitter forms an upper boundary to the SNR. The jitter, a slight difference in time apart from the exact sampling point, causes a phase shift, thus lowering the SNR. The following formula assumes that the signal to be measured is a single sine signal with the frequency $f_{in}$. The root-mean-square of the jitter is given by $t_j$[34,35].

$$\text{SNR}_{\text{Jitter,dB}} = 20\log_{10}\left(2\pi \cdot f_{in} \cdot t_j\right)$$
(11)

Other noise sources have to be considered as well. The thermal noise is often independent, and the power can be calculated by:

$$P_{\text{Thermal Noise,dB}} = 10\log_{10}\left(k_B \cdot \text{BW} \cdot T\right),$$
(12)

where $k_B$ represents the Boltzmann constant, and BW is the input bandwidth. $T$ is given in Kelvin (K). For EIS measurements, pink noise, also known as 1/f or flicker noise, plays an important role, as the power of this noise increases with lower frequencies. Thus, at lower frequencies, this source of noise can be dominant. Most signals measured during an impedance measurement pass an operational amplifier, which adds pink noise to the signal[36].

Often, it is impossible to assume that the root cause of other distortions is independent. Thus, the different sources can not be represented as additive noise.

**Signal processing and the signal-to-noise ratio.** The SNR can be improved by filtering and processing the signals, such as the voltage and the current. In real-world applications, an input filter is often placed on the PCB. In addition, the wires connecting the battery to the measurement board also serve as filters. This limits the bandwidth of the input signal. By carefully dimensioning the input filter, the input bandwidth BW of the ADC can be reduced. The filter's cut-off frequency should be below the Nyquist frequency $f_N$ determined by the sampling rate $f_N = \frac{f_s}{2}$, to improve the SNR[37].

$$\text{SNR}_{\text{Process Gain, dB}} = \underbrace{10\log_{10}\left(\frac{f_s}{2\cdot\text{BW}}\right)}_{\text{BW}=\frac{f_s}{2}=0\,\text{dB}}$$
(13)

Averaging signals can also improve the SNR, especially if the noise is uncorrelated to the signal. The SNR can be improved with each repetition (avg) of the measurements[38]. This averaging can be done in the frequency domain for EIS measurements, thus after the Fourier Transform.

$$\text{SNR}_{\text{Rolling Average Gain,dB}} = 10\log_{10}\left(\text{avg}\right)$$
(14)

The fast Fourier transform (FFT) itself can also improve the SNR. The size $M$ of the FFT results in the improvement of the SNR[37].

$$\text{SNR}_{\text{FFT Gain,dB}} = 10\log_{10}\left(\frac{M}{2}\right)$$
(15)

If all processing steps are combined, the noise floor is given by:

$$\begin{aligned}
\text{Noise floor}_{\text{dB}} = \min\Big\{ &\text{SNR}_{\text{Number Format, dB}}, \text{SNR}_{\text{ADC}_{\text{linear}}, \text{dB}}, \text{SNR}_{\text{Pink Noise, dB}}, \\
&\text{SNR}_{\text{Jitter, dB}}, \text{SNR}_{\text{Thermal Noise, dB}}, \cdots \Big\} \\
&+ \underbrace{10\log_{10}\left(\frac{f_s}{2\cdot\text{BW}}\right)}_{\text{Sampling Rate and Filtering}} + \underbrace{10\log_{10}\left(\frac{M}{2}\right)}_{\text{FFT}} + \underbrace{10\log_{10}\left(\text{avg}\right)}_{\text{Averaging}}.
\end{aligned}$$
(16)

Assuming that $\text{SNR}_{\text{ADC}_{\text{linear}},\text{dB}}$ is the minimum for a single sine signal, Equations (10) and (16) lead to:

$$\begin{aligned}
\text{SNR}_{\text{dB}} = &\underbrace{\underbrace{20\log_{10}\left(2^N\right)}_{\text{Quantization Noise}} + \underbrace{20\log_{10}\left(\sqrt{\frac{3}{2}}\right)}_{1.76\,\text{dB}} + \underbrace{20\log_{10}\left(\frac{A_{p-p}}{A_{FSR}}\right)}_{\substack{A_{p-p}=A_{FSR}= \\ 0\,\text{dB}}}}_{\text{Sine Signal}} \\
&+ \underbrace{10\log_{10}\left(\frac{f_s}{2\cdot\text{BW}}\right)}_{\text{Sampling Rate and Filtering}} + \underbrace{10\log_{10}\left(\frac{M}{2}\right)}_{\text{FFT}} + \underbrace{10\log_{10}\left(\text{avg}\right)}_{\text{Averaging}}.
\end{aligned}$$
(17)

The relationships lead to the fact that a quadrupling of the FFT size $M$ corresponds to an additional resolution of 1 bit. This also applies to the sampling rate $f_s$ and the quantity of how often (avg) the mean value is calculated. If the input filter reduces the bandwidth BW by one quarter, this is also equivalent to 1 bit ADC resolution.

$$M\cdot 4 \Leftrightarrow f_s\cdot 4 \Leftrightarrow \text{avg}\cdot 4 \Leftrightarrow \text{BW}\cdot\frac{1}{4} \Leftrightarrow N+1$$
(18)

## Results

The European Union-funded project EVERLASTING aimed to demonstrate the balancing resistor-based online EIS in an electric van. Therefore, a modular approach was chosen, as other project partners had further sensors. This led to the decision to put a second balancing resistor parallel to the standard balancing system of the BMS. Figure 2 gives a

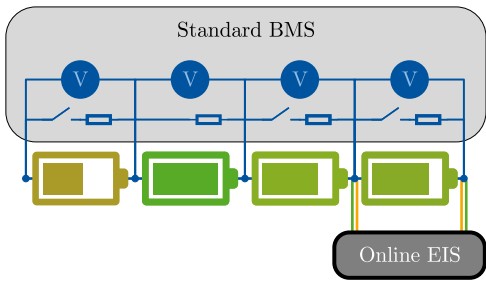

**Fig. 2 | Schematic illustration of the integration into the demonstrator.** The developed online electrochemical impedance spectroscopy (EIS) device is connected in parallel to the main battery management system (BMS). The board is connected by a four-terminal pair connection to a battery. This ensures that the voltage (V) measurement is independent of the current path.

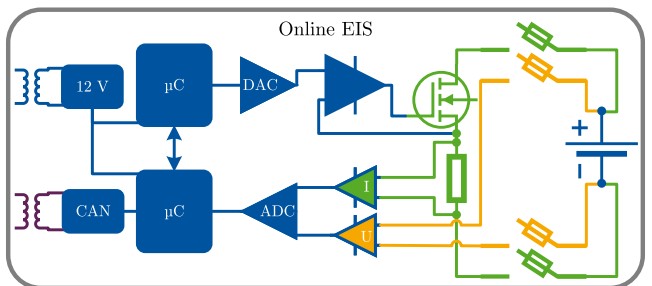

**Fig. 3 | Structural schematic of the developed online electrochemical impedance spectroscopy (EIS) board.** Communication and power supply connectors on the left are isolated from the right side of the printed circuit board (PCB). Two microcontroller (μC) are used, one for the excitation by the digital-to-analog converter (DAC) and one for the measurements of the analog-to-digital converter (ADC). The balancing current is modulated with a voltage-to-current converter setup using an operational amplifier for excitation. The galvanically isolated 12 V power supply and the connection for the controller area network (CAN) connector are located on the left.

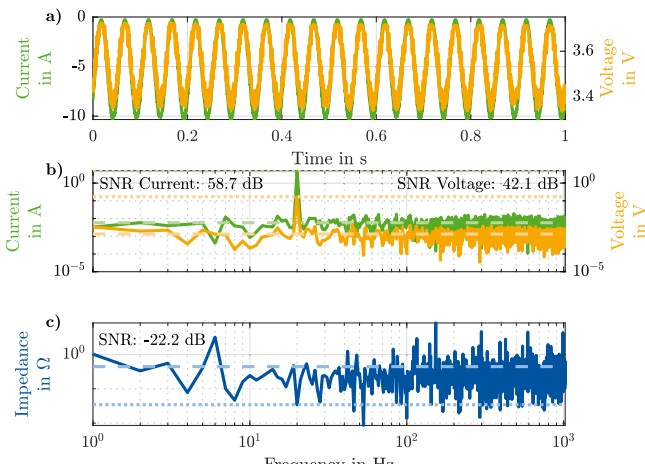

**Fig. 4 | Raw data of a 20 Hz online electrochemical impedance spectroscopy (EIS) measurement.** The voltage and current in the time domain (**a**) and the voltage and current in the frequency domain (**b**) are calculated by the battery management system (BMS). The impedance spectra (**c**) is calculated during post-processing. The signal-to-noise ratio (SNR) of the individual signals is calculated as the difference between the mean of the signal (dotted lines) and the mean of the noise (dashed lines).

polylactic acid. Through the CAN bus, both the raw data in the time domain of the current and the voltage and the onboard calculated online EIS can be transmitted.

## Measurement results

Selecting the appropriate excitation frequencies is essential, as it directly influences the accuracy and efficiency of measurements. Lower frequencies lead to longer measurement durations, making them impractical for real-time applications like fast charging. Conversely, higher frequencies require more sophisticated and costly hardware, limiting their feasibility for the mass market. Moreover, when frequencies are measured one after the other, the total measurement time scales with the number of frequencies, further complicating the process.

The 18650-LG-Chem-MJ1 battery is sensitive to aging in the medium to lower frequency range[41]. As a compromise, the final frequencies used for excitation are $f_{\text{excited}} = 2, 5, 10, 20, 40, 80, 160, 320, 640$ Hz. In combination with the sampling rates of $f_{s_{\text{ADC}}} = 2048$ Hz, $f_{s_{\text{DAC}}} = 10.24$ kHz, and the FFT size of $M = 2048$ bit, the excitation frequencies hit exactly the FFT bins.

Each frequency is excited for a minimum duration of 1.1 s. The evaluation period of the ADC is always exactly 1 s. As only complete periods are excited the 2 Hz excitation for example takes $\lceil 1.1 \, \text{s} \cdot 2 \, \text{Hz} \rceil \cdot \frac{1}{2 \, \text{Hz}} = 3 \cdot \frac{1}{2 \, \text{Hz}} = 1.5 \, \text{s} \geq 1.1 \, \text{s}$. This leads to 10.4 s excitation time for one complete sweep of all frequencies. The overall excitation time could be reduced according to Küpfmüller's uncertainty principle to $\frac{1}{2 \, \text{Hz}} = 0.5 \, \text{s}$ by using broadband excitation signals. As derived in Equation (9), broadband signals always lead to a lower SNR. Optimizing the number of frequencies excited at the same time until the SNR gets too low is not done here. The 10.4 s excitation is long enough that the system could change during the measurement and thus being not time-invariant anymore. Violating the linear time-invariant criteria is crucial and could lead to misinterpretation of the measured impedance. For example, the Kramers–Kronig relations can be applied to the measurement before interpretation to filter out measurements where the linear time-invariant criteria is violated.

Figure 4 shows the raw data of a $f_{\text{excited}} = 20$ Hz measurement. The measurement is done with one 18650-LG-Chem-MJ1 at room temperature $\approx 23 \, ^\circ$C with an excitation current of $I_{20 \, \text{Hz}} = 10 \, \text{A}_{\text{p-p}}$. The SNR is calculated as the ratio of the squared absolute value of excited frequency over the squared absolute value of the standard deviation of all other frequencies, excluding the direct current value.

schematic impression of the setup. The hardware was not optimized for weight, volume, or cost. The presented measurement results are from laboratory tests. The identical software and hardware were also operated in the van.

## Hardware setup of the demonstrator

The developed device is integrated into a 42.9 KW h battery pack of a VOLTIA electric van parallel to a standard BMS. In the pack, twenty 18650-LG-Chem-MJ1 cells are connected in parallel[39]. Thus, the theoretical 1 kHz resistance is $|\underline{Z}(1 \, \text{kHz})| = \frac{30 \, \text{m}\Omega}{20} = 1.5 \, \text{m}\Omega$[40]. An excitation current of:

$$|I(1 \, \text{kHz})| = \frac{10 \, \text{mV}_{\text{p-p}}}{1.5 \, \text{m}\Omega} \approx 6.67 \, \text{A}_{\text{p-p}} \tag{19}$$

is targeted to be comparable to lab measurements. Thus, a resistance of $R_{\text{bal}} \leq \frac{3 \, \text{V}}{6.67 \, \text{A}_{\text{p-p}}} \approx 450 \, \text{m}\Omega$ is necessary. As shown in Fig. 3 a Voltage-to-Current converter setup using an operational amplifier is chosen for excitation. A pulse-width-modulation variant is also possible. For further modularization, two LPC1769 Cortex-M3 microcontrollers by NXP are used. Also, a 4-terminal-sensing is implemented. The digital-to-analog converter is the AD5541A by Analog Devices. The ADC is the ADS8568 by Texas Instruments. The connectors of the controller area network-Bus and the 12 V supply are galvanostatic isolated from the PCB's high-voltage side. The final resistor is the PWR247T-100-R200F by Bourns, with 200 mΩ resistance. Furthermore, large heat sinks 6398BG by BOYD are attached to the metal-oxide-semiconductor field effect transistor and the resistor to keep the temperature below 50 °C. In this research project, the housing and mounting are partially made with a 3D printer using

In Fig. 4a, the raw data in the time domain of the excitation current of the $I_{20\,Hz} = 10$ A$_{P-P}$ signal and the voltage response of the battery is shown. The BMS calculated the frequency domain values in Fig. 4b. The SNR values were calculated during post-processing on a computer. The amplitudes of the measured frequency are plotted with dotted lines, and the means of the noises are plotted with dashed lines. The current's amplitude at 20 Hz (half of the peak-to-peak amplitude) is 4957.56 mA, and the voltage has an amplitude of 170.75 mV. The noise levels are at 5.78 mA and 1.33 mV, resulting in an SNR of 58.7 dB and 42.1 dB, respectively.

In the BMS, only the impedance of the excited frequency is evaluated and communicated via the controller area network. In the post-processing, the impedance of the whole spectrum was calculated based on the logged raw data. Figure 4c shows that this results in a negative SNR for the impedance. The signal of the excited frequency is 34.44 mΩ, whereas the mean is 444.61 mΩ.

Further experiments are carried out to investigate and compare the demonstrator's performance. Figure 5 shows the testbed of the experiment (a) and the test setup of a reference measurement (b). The experiment was conducted with a single fully charged 18650-LG-Chem-MJ1 battery cell. A new measurement for all frequencies was taken every minute. The measurement starts were triggered by a controller area network message sent by an external microcontroller. This slowly discharged the cell. A thermal camera tracked the temperature, and all data were logged into a laptop. The EIS device EISmeter and the battery cycler MCT 200-06-3 ME ME from Digatron Power Electronics together with a temperature chamber BINDER MK 53 are used for the reference measurement.

Figure 6 plots the remaining electric charge of the battery on the *x*-axis and the impedance change on the *y*-axis of both the experiment and the reference measurement. Figure 6a, c shows the results of the experiment, and (b) and (d) are the reference measurement results. The impedance of the cell is relatively similar at higher and lower SoCs. The impact of the self-heating of the battery cell at the beginning of the experiment is not considered here. Compared to the phase, the absolute value increases more to lower SoCs than decreases at higher SoCs. Nyquist diagrams can be found in Fig. 6e, f. The phase difference is more visible here.

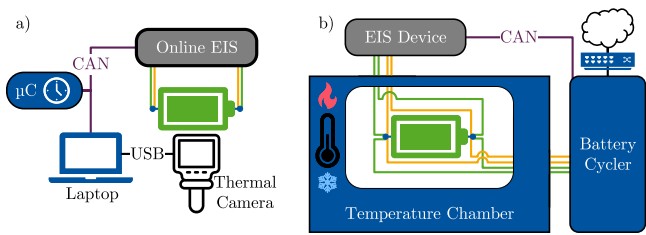

**Fig. 5 | Testbed of the experiment. a** using a microcontroller (μC) to trigger trough controller area network (CAN) impedance spectroscopy (EIS) measurements periodically. A thermal camera captures the temperature, and the data is collected on a laptop. For reference measurements (**b**) a battery is placed inside a temperature chamber and discharged with a battery cycler. A CAN bus connects the EIS device to the cycler. The data is stored in the cloud.

**Fig. 6 | Online electrochemical impedance spectroscopy (EIS) measurement results.** Phase (**a**) and the absolute value (**c**) of the impedance of a battery over different state-of-charges (SoCs) during the experiment while (**b**) and (**d**) showing the results of the reference measurement. The measurement frequencies range from 2 to 640 Hz. The absolute value of the impedance is lowest for medium SoCs. The online EIS results of (**a**) and (**c**) are calculated in the battery management system (BMS). Nyquist diagram during a discharge of a battery with a balancing resistor for online EIS measurements (**e**) and reference measurements (**f**). The excitation frequencies range from 2 Hz to 640 Hz in (**e**) and from 10 mHz to 6 kHz in (**f**), whereby only the range from 2 Hz to 640 Hz is colored.

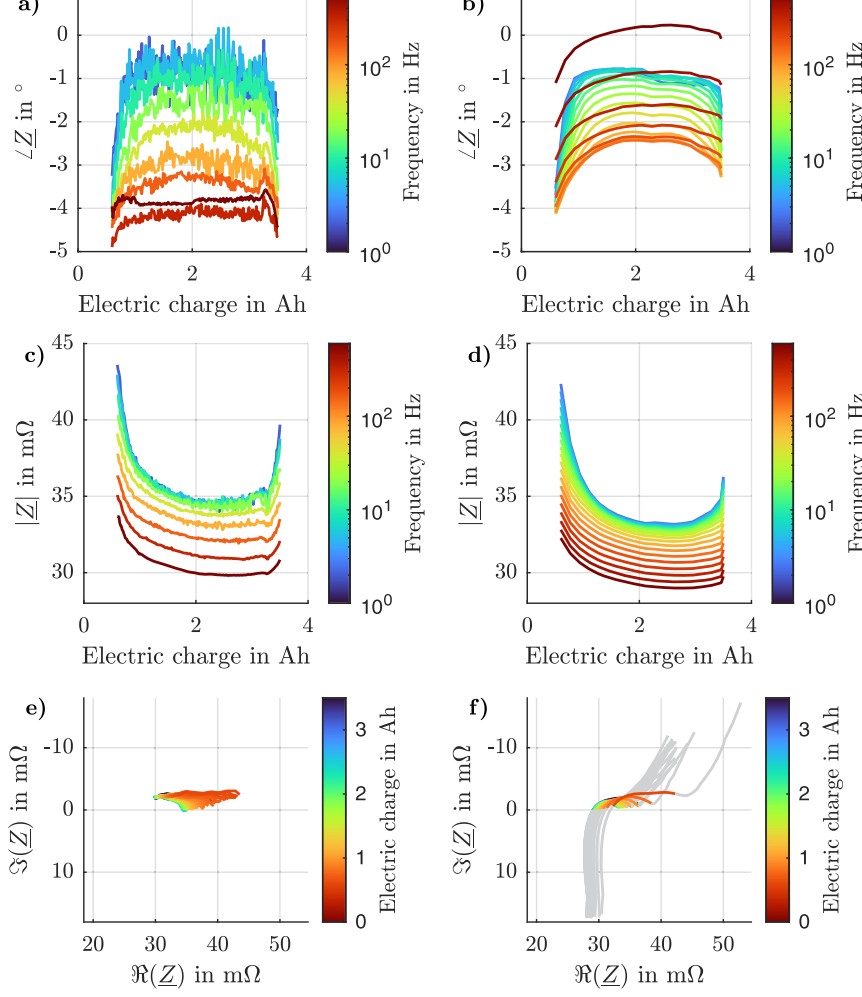

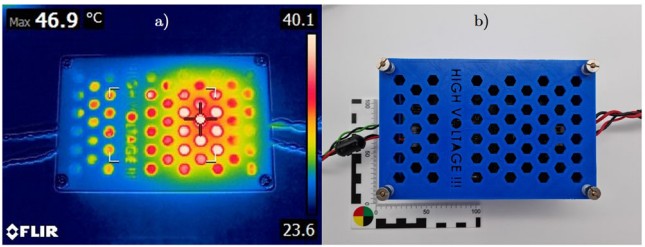

**Fig. 7 | Photos of the developed excitation board.** Thermal imaging (**a**) and picture (**b**) of the online electrochemical impedance spectroscopy (EIS) board in a polylactic acid (PLA) housing during a discharge test. The peak temperature is 46.9 °C. The size of the developed printed circuit board (PCB) is 100 × 160 mm (Eurocard format). **a**, **b** have the same scale, which is photographed in (**b**).

The device's temperature was observed with the thermal camera during the experiment. The peak temperature of the board was about 47 °C. The frame of the video is shown in Fig. 7.

## Discussion

For the before-mentioned battery lithium-iron-phosphate CATL 271 Ah[16] with $|\underline{Z}(1 \text{ kHz})| = 0.14 \pm 0.05 \text{ m}\Omega$, a current of $I = \frac{10 \text{ mV}_{\text{P-p}}}{0.14 \text{ m}\Omega} = 71.43 \text{ A}_{\text{p-p}} = 0.26 \text{ C}_{\text{p-p}}$ would be necessary to achieve a voltage response of $10 \text{ mV}_{\text{P-p}}$. Even though EIS measurements in the lab often aim at a voltage response of 10 mV, lower voltages are possible if the SNR of the voltage measurements allows it.

The impedance data shown in Fig. 6 aligns with the findings from ref. [42]. Although the measurements from the developed demonstrator vary from those obtained using commercial laboratory equipment, the absolute values are closely matched. However, the noise of the absolute values and phases increases at lower frequencies. The phase measurements show greater deviation at higher frequencies. While the EIS data is not interpreted in this study, future steps could include fitting and modeling using, for example, an equivalent circuit model, which would enable not just measurements but also online diagnostics.

Generally, it is possible to embed online EIS into a BMS by accepting high temperatures on the PCB. Increasing the capacity of the battery results in decreasing the impedance. If, instead of 1 cell, 20 cells in parallel are measured, the voltage SNR drops by about 26 dB. In Fig. 6, the SNR decreases with lower frequencies. This could indicate that pink noise affects the measurements. Extending Equations (16) and (17) by additional impacts is of interest, as this would greatly improve the initial system design.

In the case of active excitation, the exited current could also be calculated instead of being measured. This includes the benefit that the SNR of the impedance itself only depends on the SNR of the voltage. The online impedance SNR calculation approach by ref. [4] could then be adapted to this. This would allow to rate the precision of the online EIS in terms of SNR during the measurement.

Sizing the balancing resistor for absolute extreme cases, as it was done in Equation (4) and following, leads to too low resistance values. As the self-discharge rate is higher with higher voltages, a battery system somewhat equalizes itself. More investigations of the self-discharge currents are necessary to model this further and include this aspect in the system design phase. The longer balancing is possible in the application, the lower the balancing current can be. Lower balancing currents require higher resistance and lead to reduced size and heat. If the system is not in use, waking up the BMS for balancing might not be optimal. Balancing all the time during operation, on the other hand, would increase the balancing time. Cell-dependent self-discharge estimations are necessary to enable this. Furthermore, the voltage drop during balancing has to be compensated.

For the comparison of balancing resistors standardization is necessary. At least the battery capacity should be included. The implemented balancing strategy and the voltage level of the used chemistry affect the effective balancing current for a given resistor. Thus, from a reverse engineering point of view, a standardized normalization might not be helpful.

## Conclusion

Online EIS analyzes a chemical system during operation by its complex impedance. To extract the impedance, an excitation is necessary. The balancing resistor in the battery system can excite the battery, e.g., with sinusoidal signals. Current and voltage measurements with a decent SNR are necessary for the impedance measurement. The used hardware and environmental conditions affect the SNR, such as thermal noise. Advanced signal processing can improve the SNR. It is necessary to start with a suitable number format. FFT size, averaging factors, sampling rate, and filter bandwidth can compensate for analog resolution, as summarized in Equation (18).

Passive balancing with a resistor as a thermal sink of energy can be used to equalize differences in voltage or SoCs in a battery system. The difference in self-discharge rates of batteries drives the sizing of the resistance of the resistor. Limitations can come from the time of balancing and peak temperatures allowed on the PCB. A proof of concept was carried out, showing that online EIS with a balancing resistor is possible. Key challenges are the heat generated during balancing and the precision of the voltage measurement. The developed hardware, as such, is not optimized to be directly integrated into an application. Measurement results from a battery discharge via balancing EIS are carried out and aligned with other published data and reference measurements.

In conclusion, we analyzed the sizing of active dissipative balancing systems. If individual cells or modules of a battery system are to be replaceable, the difference in the self-discharge of the batteries varies more after replacing individual cells or modules. Therefore, the balancing current must increase, resulting in a higher amplitude and, thus a higher SNR. The higher amplitude could be used for online EIS. The measured impedance of a battery can then be used for temperature, SoC, state-of-health, or other estimations. Advanced signal processing techniques have the potential to eliminate additional implementation costs. This leads to safer, more reliable and sustainable battery systems.

## Data availability

The data gathered in the experimental work of this study, supporting the findings of this work are available from the corresponding author upon reasonable request.

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

## Acknowledgements

 This research received funding from the European Union (EU) in the project EVERLASTING (funding code: 713771), from the Deutsche Forschungsgemeinschaft (DFG, German Research Foundation) in the project mobilEM (funding code: GRK 1856), and the Federal Ministry of Education and Research (BMBF) in the project OSLiB (funding code: 03XP0330C).

## Author contributions

The work of the authors splits as follows: Alexander Blömeke: conceptualization, data curation, formal analysis, investigation, methodology, project administration, resources, software, supervision, validation, visualization, writing - original draft preparation. Hendrik Zappen: conceptualization, writing - review & editing. florian ringbeck: funding acquisition, project administration, resources, supervision, writing - review &

editing. Fabian Frie: data curation, investigation, funding acquisition, project administration, resources. David Wasylowski: funding acquisition, project administration, resources, supervision, writing - review & editing. Dirk Uwe Sauer: funding acquisition, project administration, resources, supervision, writing - review & editing.

## Funding

## Competing interests
Weihan Li is a guest editor for the Communications Engineering Collection on Battery Management Systems for Vehicle Electrification. He is also a colleague of some of the authors of this contribution. However, he was not involved in the editorial review of, nor the decision to publish this article. The authors declare that they have no further competing interests.
