## [Peer Review File · Communications Engineering]

Reviewers' comments:

Reviewer #1 (Remarks to the Author):

The reviewer would like to give some comments for improving the paper as follows:

- Revise Fig. 1: It should depict a more specific concept, and the caption should describe the figure not contents of the paper.
- On page 5, the definition of CN is not clear.
- Re-define R_{bal} , using a different symbol due to its unit being different from ohms.
- Please show an overview of the testbed setup, including the test board and battery pack.
- In Fig. 4, why is the CAN connected to the transformer?
- The selected frequencies of EIS is insufficient
- Based on the EIS results, what are the equivalent circuit parameters?
- Compare EIS results from the demonstrator to the available EIS data.

Reviewer #2 (Remarks to the Author):

The paper has certain innovation and application value. The following issues need to be addressed before publication:

The EIS measurement results need to be verified, which is very crucial.

2. It is best to have some quantitative results in the abstract.

3. How to achieve full frequency measurement in practical applications?

4. The measurement results of EIS do not seem ideal, how can it be applied?

Reviewer #1 (Remarks to the Author):

The reviewer would like to give some comments for improving the paper as follows:

- Revise Fig. 1: It should depict a more specific concept, and the caption should describe the figure not contents of the paper.
 - You're right. This figure was not scientific enough. I deleted it, as it didn't give specific information.
- On page 5, the definition of CN is not clear.
 - Thanks. Added it.
- Re-define R_{bal} , using a different symbol due to its unit being different from ohms.
 - Using now the prime operator, as this is used similarly in the field of feature scaling.
- Please show an overview of the testbed setup, including the test board and battery pack.
 - In 4.2. the testbed is shown and explained in detail. Furthermore, the setup of reference measurements is shown and described.
- In Fig. 4, why is the CAN connected to the transformer?
 - I rewrote the description, as it didn't become clear that the connectors of the 12 V supply and the CAN interface are galvanostatic isolated from the battery.
- The selected frequencies of EIS is insufficient
 - I'm now describing why those frequencies were selected. These frequencies are sensitive to ageing, but also the SoC. In an application, like a vehicle, and especially during operation a complete spectra will never be measured. The goal is not have compatible results to lab measurements, but to have necessary information to track e.g. the SoC better.
- Based on the EIS results, what are the equivalent circuit parameters?
 - I thought about adding ECM fits during writing. In the end, I decided not to add ECM parameters, as this paper is already very packed with different information and modelling/ fitting would open up a new topic. If you are okay with that, I would be happy if you think it's okay that I don't take up this topic here. A more extended discussion is now added to compare the EIS results, where I mention that, e.g., ECM fitting of the results could be used to track the changes of a battery.
- Compare EIS results from the demonstrator to the available EIS data.
 - Thank you very much. A very good point. A colleague had thankfully already done characterization measurements back then and could now add the reference measurements relatively simple. The differences are analysed in the discussion section. Thank you again. Very important point that I had missed to add. As the measurements are done and interpreted by my colleague, I added him as a co-author.

Reviewer #2 (Remarks to the Author):

The paper has certain innovation and application value. The following issues need to be addressed before publication:

- The EIS measurement results need to be verified, which is very crucial.
 - Thank you very much. A very good point. A colleague had thankfully already done characterization measurements back then and could now add the reference measurements relatively simple. The differences are analysed in the discussion section. Thank you again. Very important point that I had missed to add. As the measurements are done and interpreted by my colleague, I added him as a co-author.
- It is best to have some quantitative results in the abstract.
 - I've reformulated it to focus on Formula 18, which helps BMS developers choose/compensate for sensor resolution by signal processing.
- How to achieve full frequency measurement in practical applications?
 - I now focus more on explaining why I chose these frequencies. Furthermore, I added to the discussion that the frequency set has to be selected depending on the battery cells and the SoX that should be improved.
In practical applications, I end up at the tradeoff between how quick a measurement should be (the fewer frequencies, the quicker) and what should be measured (e.g., SoC, SoH, or Temperature). I added an explanation that choosing correct frequencies and analyzing the impedance (e.g., with ECMs) is not the focus of this paper. I hope you agree that these questions are essential but could become a separate paper in terms of length.
- The measurement results of EIS do not seem ideal, how can it be applied?
 - I hope the comparison with the laboratory measurements makes it clear that the measurements only cover a section of the spectrum and, therefore, seem less strange. I am now addressing this point more in the discussion.

Reviewers' comments:

Reviewer #2 (Remarks to the Author):

The author has already solved the problem I am concerned about very well. I recommend publishing this work.

Reviewer #3 (Remarks to the Author):

In the manuscript entitled "Balancing Resistor-based Online Electrochemical Impedance Spectroscopy: Opportunities and Limitations" the authors want to explore the feasibility of using the resistors of the active dissipative balancing system in the battery management systems (BMS) to stimulate battery cells for EIS measurement. Potential and issues of integrate this approach in the BMS are investigated.

The manuscript is quite well written regarding the flow of reasoning, the outline and the language. I would point out some notes.

The authors present the method as an ONLINE method to perform the EIS, but the "frequency sweep" is a well known technique to be a slow technique to measure the impedance. Broadband excitation is the appropriate technique to perform a real-time diagnosis of the battery, as named by the authors and already proposed in the literature. Moreover, long time measurement could imply violation of the stationary behavior of the battery, since during the operation conditions the internal state of the cell can change significantly within seconds. How long is the complete excitation pattern?

In my opinion Fig. A2 is a crucial result of the work and it should not be in the APPENDIX section. The frequency range considered in a) should be highlighted in b).

Reviewer #2 (Remarks to the Author):

The author has already solved the problem I am concerned about very well. I recommend publishing this work.

Reviewer #3 (Remarks to the Author):

In the manuscript entitled "Balancing Resistor-based Online Electrochemical Impedance Spectroscopy: Opportunities and Limitations" the authors want to explore the feasibility of using the resistors of the active dissipative balancing system in the battery management systems (BMS) to stimulate battery cells for EIS measurement. Potential and issues of integrate this approach in the BMS are investigated.

The manuscript is quite well written regarding the flow of reasoning, the outline and the language. I would point out some notes.

The authors present the method as an ONLINE method to perform the EIS, but the "frequency sweep" is a well known technique to be a slow technique to measure the impedance. Broadband excitation is the appropriate technique to perform a real-time diagnosis of the battery, as named by the authors and already proposed in the literature. Moreover, long time measurement could imply violation of the stationary behavior of the battery, since during the operation conditions the internal state of the cell can change significantly within seconds. How long is the complete excitation pattern?

- Thank you very much. I have now addressed and explained the problem more directly in section 4.2. The aspect of the LTI system, in particular, was not sufficiently discussed in the previous version.

In my opinion Fig. A2 is a crucial result of the work and it should not be in the APPENDIX section. The frequency range considered in a) should be highlighted in b).

- The highlighting helped a lot. It's a great idea. I shifted it to Figure 6.